# Behavioral Analysis and Individual Tracking Based on Kalman Filter: Application in an Urban Environment

**DOI:** 10.3390/s21217234

**Published:** 2021-10-30

**Authors:** Amaury Auguste, Wissam Kaddah, Marwa Elbouz, Ghislain Oudinet, Ayman Alfalou

**Affiliations:** 1L@bISEN, Equipe LSL, Yncrea Ouest, 20 Rue Cuirasse Bretagne, 29200 Brest, France; amaury.auguste@yncrea.fr (A.A.); wissam.kaddah@isen-ouest.yncrea.fr (W.K.); marwa.el-bouz@isen-ouest.yncrea.fr (M.E.); 2ISEN Yncréa Méditerranée, Pl. Georges Pompidou, 83000 Toulon, France; ghislain.oudinet@yncrea.fr

**Keywords:** Kalman filters, video tracking, behavioral analysis, YOLO, anonymity, data analysis, clustering

## Abstract

In order to improve behavioral analysis systems in urban environments, this paper proposes, using data extracted from video surveillance cameras, a tracking method through two approaches. The first approach consists in comparing the position of people between two images of a video and to perform tracking by proximity. The second method using Kalman filters is based on the anticipation of the position of an individual in the upcoming image. The use of this method proves to be more efficient as it allows continuing a detection even when people cross each other or when they pass behind obstacles. The use of Kalman filters in this domain provides a new approach to obtain reliable tracking and information on speed and trajectory variations. The proposed method is innovative in the way the tracking is performed and the results are exploited. Experiments were conducted in a real situation and showed that the use of some elements of the first method could be reused to integrate a notion of distance in the method based on the Kalman filter and thus improve the latter both in tracking and in detecting of abnormal behavior. This article deals with the functioning of the two methods as well as the results obtained with the same scenarios. The experimentation concludes through concrete results that the Kalman filter method is more efficient than the proximity method alone. A sample result is available online for two of the seven videos used in this article (accessed on 19 July 2021).

## 1. Introduction

The study of behavior can take several forms. In the computer field, the methods most often put forward are data monitoring and analysis through “data mining” to analyze, for example, consumers’ needs [1], but also to prevent terrorist acts [2]. However, since the amount of data to be processed is often large, these analyses take time and do not allow prevention of a spontaneous act. Another method that is increasingly used today is behavioral analysis of real-time data using, for example, video surveillance cameras. These analyses can be simple, for example, by monitoring by a human operator, or much more complex thanks to video analysis. The subject of this article is based on the improvement of these video surveillance systems by adding the possibility of individualized tracking. The need for individualized tracking arose from research conducted as part of a project to secure urban spaces in the city of Nice, France [3]. This project aimed at improving security in urban spaces and allowed to carry out research in several fields such as the use of artificial intelligence for the detection of intrusions or abnormal behavior [4], the detection of morphology for the differentiation of an adult or a child [5], or facial recognition based on neural networks and correlation techniques [6]. During this research, it has been put forward that the identification and tracking of people visible from a video surveillance camera could improve the detection of dangerous and abnormal behavior. This article therefore deals with the creation of an individualized tracking system, as a follow-up to the work carried out in the framework of the security project. The objective is to be able to make more specific analyses during panic movements, but also to have more information on an individual and his trajectories.

The desired tracking can be done in several ways, the first would be purely oriented on video processing. The most used methods today are based on the analysis of the video and thus of groups of pixels evolving in a short time to detect people and sometimes to track them [7]. In general, it appears that the part related to detection is powerful and that the quality of the latter will depend on the tool or technology used such as R-CNN (Region Based Convolutional Neural Networks) [8] or YOLO (You Only Look Once) [9] which are both dependent on artificial intelligence and which can detect a multitude of objects once they have been sufficiently trained. In this paper we will use YOLO detection which is more efficient than CNN, R-CNN, or Faster R-CNN [10].

For tracking, the results are sometimes more questionable due to the number of parameters that can change (frame rate, image resolution, camera exposure…). In general, a good quality of the video (resolution and frequency) and stability of the camera will allow the best possible detections [11]. Therefore, the method proposed in this paper is based on individual detections in videos, but also on tracking algorithms that are not dependent on a video system. However, some methods such as “Kanade-Lucas-Tomasi feature tracker” [12] allow tracking points or shapes position in a video. This method can for example be used in addition to the “Harris corner detector” [13] to track multiple elements. This article proposes a different tracking approach whose primary objective is to respect complete anonymity when analyzing the trajectory. To do so, no video data will be transmitted to perform the analysis.

The initial observation is that detection systems have become sufficiently powerful to give us the position of individuals in each frame of a video as an anonymous point cloud. It is therefore possible to do our tracking on these sets of point clouds without needing to use any other image of the video. Our method is therefore based on two main axes, namely video detection and the tracking of the evolution of points in space as a function of time.

Regarding the detection, several methods can be used to detect elements in an image. Today, the methods that are most often used are those based on AI and more particularly on “deep learning”, the use of YOLO mentioned above. The use of one of these methods allows us to generate a point cloud for each image of the video where each point represents the center of the person detected and the position of the point in the cloud will represent the position of the person in the image. Extracting a real situation into a point cloud usually has an advantage, as only the necessary information is kept, and the processing can then be much more targeted [14].

For tracking, several methods are possible, the first being algorithmically simple with distance comparisons within point clouds and probable pairings using only the known N and N-1 image data. This technique is for example used in some clustering algorithms such as DBSCAN [15]. In a second step, another method based on Kalman filters [16,17] would allow using the known data in N-1, but also would keep a trace of the data in N-X to predict how the point cloud could evolve in N.

The use of these methods, and more particularly the one using Kalman filters in the field of tracking applied to video surveillance, makes it possible to set up an instantaneous tracking system and thus a continuous identification of the persons present in the video flow. This method allows to obtain precise information not only on the position of an individual in the video stream, but also on the trajectory and the future position of this individual. The novelty and the finality of this method reside in an optimized algorithmic implementation in order to allow, through the detection of abnormal trajectory, to define if it is a simple loss of track and in which case to correct the error or if it is indeed an abnormal behavior presenting a variation of speed or unusual trajectory. Such a use of this tracking brings to the current detection methods the possibility to determine abnormal behaviors and to have a tracking that is both efficient and integrates speed and instantaneous trajectory information.

This article is therefore presented in several parts. The first part presents a rudimentary tracking method based on simple comparison rules. This method allowing good results will, however, be compared to another method based on Kalman filters that is more complex. Finally, the two methods will be compared on the same data set to quantify the real efficiency of the two methods.

## 2. Database

To validate the functioning of the algorithms proposed in this article, we had to compose a database. To do this, we can use videos from existing fixed cameras, but we preferred to make our own set of videos to have a greater flexibility in the filmed scenarios and thus to make stagings of escape, gathering, and dispersion… The shooting of these videos was done using an “OM-D EM-5 mk II” camera capable of filming in 4:2:0 8b, H.264, BT-709, 29.97 fps set at the same level as a city security camera. The filmed area is surrounded in red in Figure 1 and is located outside the laboratory in the center of Toulon, France.

It is necessary to have a good quality when we film our scenes, because the algorithm of detection of an individual will not be able to detect an individual if it is too blurred. We have therefore chosen to use these resolutions which are standard today. The study does not deal with the detection part, so we chose to get closer to the specifications of the cameras used by the city of Nice. It should be noted that the resolution of the camera will have an impact on the detection part in the sense that the detection will be much more effective on an image of the best possible quality.

Our experiments are based on seven videos among about twenty that were initially shot. They were selected for their characteristics presented in Table 1, which are:-the number of passages behind an obstacle;-the number of people who cross each other;-the number of track mergers (due to the detection algorithm);-the total number of individuals;-the duration of the video.

In order not to have to process the videos for each test, all the videos have been processed beforehand and the detections have been saved in a JSON file. The data is labeled with all the necessary information to replay the videos in real time.

In the rest of this article, we will consider that the videos have been processed beforehand and that we therefore have a set of detection replay. The detection replay allows us to simulate the evolution of a point cloud in a 2D environment, which allows us to detach ourselves completely from the video, which is not the object of the study. Indeed, in the rest of this document, we will consider that the video processing was carried out by any algorithm, but we will admit that each image of the processed video allowed the generation of a set of points forming a point cloud. We thus have a set of point clouds evolving through time. This set of point clouds has been saved in a JSON file with the following structure:{Timecode: int, individuals: [{x: int, y: int, width: int, heigth: int}, …]}(1)

Each point cloud contains a number of points corresponding to the number of individuals detected in the image. These points are defined by four characteristics: their abscissa positions on the image, their ordinate positions on the image, the height of the detection on the image, and the width of the detection on the image. The points are grouped in a table whose size varies according to the number of individuals detected in the video and to which is associated a Timecode. The Timecode value is an integer whose unit is the millisecond, all the other values of position and size being taken directly from the video are in pixels.

In this article, we will discuss how this point cloud evolution has been processed through two methods. All the processing done on these points was done and coded in Python3 under Windows with the Visual Studio code IDE. No proprietary library was used, and the matrix calculations were performed using Numpy and the calculations related to the Kalman filter were programmed in the laboratory. The results obtained after the different treatments can be reused by a system that would assemble both the image and the results if necessary to exploit the results visually. In our case of study requiring only to raise alarms according to abnormal detection it was not necessary to reintroduce the results on a video, that is why the illustrations present in this article are given as an indication and are thus a visual transcription of the results which is not in any case necessary to obtain them.

The algorithms have been tested on all the videos to obtain the data presented in the following article. The validation of the good functioning of the algorithms will be done through the stall count. A stall is when the algorithm fails to track by losing a track, reversing two tracks, or re-identifying a wrong track. These stalls will be counted through graphs generated from the data of the algorithms, an example of which is presented in Figure 2.

In Figure 2a,c and Figure 3a,c represent the position of each individual for each frame of the video, each colored line represents the potential tracking of a person. Figure 2a and Figure 3a represent their evolutions on the abscissa axis of the camera as a function of the elapsed time. Figure 2c and Figure 3c represent their evolution on the ordinate axis as a function of the elapsed time. Figure 2b and Figure 3b are a representation of all the trajectories of the people presents in the entire video. The lines are of different colors, because each color corresponds to a known point (with an identifier) which means that the tracking has been interrupted if during the walk of an individual the line changes color. Figure 2d is the same graph as Figure 2b, but what is considered as “stall” is circled in red. The video No. 1 presented in Figure 2 is a fast dispersion of initially grouped individuals. Figure 2d shows 12 stalls for video No. 1 at 30IPS (Image Per Second) processed with the first method.

In most cases, the stalls are accounted for on the graphs as presented in Figure 2b, but if the number of tracks is too large, it is sometimes necessary to use the graphs as presented in Figure 2a,c. Figure 3 shows how the same stall is identified on the 3 types of graphs (a), (b), and (c) as well as a zoom on the areas where the stall occurs: (d) zoom of (a), (e) zoom of (b), and (f) zoom of (c).

To test both algorithms under the same conditions, the algorithms received the same data. To validate the robustness of the algorithms, the experimentation was also conducted by methodically removing some data, thus increasing the time between the reception of two-point clouds. The objective of this withdrawal is to see how the algorithms behave if they are fed with data by a slower or slower detection, i.e., 10, 15, 20, 25, and 30 image processing per second. Each video was processed by the two algorithms to obtain the set of point clouds labeled with an identifier for each point of each cloud received.

## 3. Tracking by Proximity

The most basic approach was to develop an algorithm whose goal is to find a correlation between two point clouds from two images following each other in a video. Indeed, as explained in the introduction, the data of the video and more particularly the positions and sizes of the persons present in each frame of the video were extracted to obtain a point cloud for each frame of the video where each point represents a person in the image to which we also attribute information on the size of the detection. The objective of this method is to search for the closest points considering that the evolution of a person between two images depends on three parameters:-the distance at which the person is from the camera;-the speed at which the person moves;-the speed at which the camera records.

We therefore wish to form pairs of points based on the proximity of individuals between two images of the video. We first evaluate the position of the points in the cloud using only their XY coordinates. However, to proceed with this evaluation, we must first know the three observable situations at the beginning:-the number of points between the two point clouds is the same;-the number of points in cloud 1 is lower than in cloud 2;-the number of points in cloud 1 is higher than in cloud 2.

Each of these cases leads to a particularity that the algorithm must be able to handle.

Let us consider the nominal case where the number of individuals remains unchanged (Figure 4), the most logical solution is to look between the two point clouds to see which points are closest and thus form pairs of points (Figure 4d).

The DBSCAN as well as many other clustering algorithms work to cluster points in a point set based on specific parameters. The main parameter used here is the distance, as our primary goal is to cluster the closest points together. The other parameter that is important is the quantity of points to group, in our case the grouping can only be done on two points maximum. We do not use the whole algorithm, we only use the part consisting in forming groups. This part is then modified so as not to form groups in the same point cloud, but to form groups between two point clouds. Any point belonging to the first cloud can only be grouped with a point belonging to a second cloud. This method works very well if the number of points between the first cloud and the second cloud remains unchanged, if the points that are present in the first cloud are also present in the second cloud and if only their position has changed. Apart from this very specific and ideal case, others exist.

In order to carry out the groupings with DBSCAN, each point of the A cloud is given a detection radius. We then simply search in the cloud B for all the points that fit into the detection circles previously established. As shown in Figure 5, in a normal use, several points can be grouped together if they are inside several detection radii. In our case, we will try to have only one point in each detection radius.

As shown in Figure 6, the proximity processing algorithm is as follows:

As explained earlier, the video stream is processed independently of analysis systems. Once processed, the result is stored as a JSON file. The first step is to convert the data in the JSON file into data that can be processed by the language used (in our case python3). If it is then the first data of the file, they are automatically integrated into the known points. The known points in our case are all the points with an identifier. If it is not the first message, the first step is to form the possible pairs between the old known points and the new received points. Three results are then possible:-The new point is associable with an old point;-The new point is not associable with an old point;-Several points are associable with an old point.

In the first case, the solution is simple, there is an old and a new point perfectly associable, the new point is thus assigned information from the old one.

In the second case, if the new point can only be associated with an old one, it may mean that either the new point is a bad detection, or that the person represented by this point has passed behind an obstacle or has left the field of vision of the camera. In both cases, the point is saved in memory to be used later. However, this storage in memory must be temporary, because if it is not just a temporary disappearance, the point must not be reassigned.

In the third case, if several new points are sociable to an old point, we simply associate the closest point to the old point and the other points are kept in memory in the same way as for the second case.

However, this solution is problematic if between two images one of the individuals has disappeared and another has appeared elsewhere in the image. Indeed, we have the same number of points between the two clouds (Figure 7a), so we can form our couples, however, the distance between some couples will be too large for it to be effectively the same person (Figure 7b).

To correct this problem, a simple solution is to set a radius around each point established through observation of the scene and the environment for several hours. Once the radius is established, if a point appears outside all the rays present in the cloud, it will most likely be a new individual, in which case we can conclude that another person has disappeared from the image. This technique follows the logic of a DBSCAN [15] clustering method that allows both grouping and identification of single individuals.

In the case where the number of points in the first cloud is less than that in the second cloud, we are faced with the disappearance of an individual. This disappearance can be due either to an exit from the area filmed by the camera or to a passage behind an obstacle (Figure 8a,b) which would prevent the camera from identifying the individual (Figure 8c). In the first case, the simplest solution is to check where the disappearance took place. If we are close to the edge of the monitored area, we will consider that the person has effectively left the field, making sure that the value of proximity to the edge used is as small as possible to ensure that the person continues to be tracked if he or she passes behind an obstacle that is close to an edge for example. In the second case, if the person disappears somewhere other than near one of the edges of the monitored area, this means that the person has most likely passed behind an obstacle and can potentially reappear. In this case, the obvious pairs must be formed, and a record kept of the last known position of the person who disappeared. Keeping track of all the positions will potentially allow the person to be recovered if he/she were to reappear. If the person is lost for too long and then reappears (Figure 8d,e), then they will be identified as a new person (Figure 8f).

In order for this method to work, we have to set a number of images during which the lost points will be kept in memory, because if we keep these points in memory for too long we could then detect a new individual that would appear at the same location as a disappearance and thus the tracking would assume that it is the same person.

Finally, in the case where the number of individuals in the first image is greater than that in the second image and we have not managed to form a pair from points that were in memory, this means that an individual has appeared in the field of view of the camera. In this case, we need to check if the appearance took place near one of the edges, because this would simply mean that there is a new individual in the image, so there will be no pair associated with the old image for this point.

In the case where the number of individuals in the first image is greater than in the second image (Figure 9), we are then in a case of appearance which may be due to the fact that a person enters the field of view of the camera or that a person was hidden behind an obstacle and is now observable again. We will then first establish all the obvious pairs between the two images using the grouping method explained previously and we then consider that the person who has just appeared is a new person to be tracked (Figure 9d).

The results show that the method works in the ideal case where the number of processed images is sufficiently large (ideally greater than 20), the field of view of the camera is sufficiently clear and the individuals present in the scene do not have intersecting trajectories. As soon as these parameters are met, we have an efficient and fast algorithm. It is efficient because the processing is only done between two clouds of points and each point will have a most probable identifier which often turns out to be the right one. The algorithm is also fast, indeed, once all the point comparisons are done, the algorithm is able to give an identifier to all the points. However, the algorithm is sensitive to the crossing of individuals and the passage behind obstacles as shown in Table 2.

Our study aims to perform tracking in an urban environment, the crossing and passing behind an obstacle is therefore regular. We must therefore find a solution that can manage these stalls.

## 4. Kalman Filter Tracking

In order to correct the method presented in the previous paragraph, it was necessary to find a way to estimate the probable evolution between two clouds of points by their distances, but also by their past behavior. Several methods, notably in the maritime domain [18,19], allow for example to estimate the trajectory of marine or even air vehicles [20,21] which have a strong inertia using Kalman filters. Since Kalman filters can be used from position and velocity data, they can also be used for airborne guidance when there is no GPS or other satellite navigation technology [22]. Kalman filters can also be used in signal processing for smoothing or estimation [23]. The field of application is therefore very wide, which is why the use of such a filter has become obvious. Such a method could allow us to make predictions on the position of future points in our point cloud, because even if the inertia of a vehicle is greater than that of a walking pedestrian, due to the difference in mass, the evolution of the latter is not random and maintains a fairly regular trajectory with smooth angles. However, it will be important to integrate the idea that a trajectory can be disturbed and potentially reveal a trajectory error or an abnormal behavior. Using Kalman filters, we are able to estimate the future trajectory of an object moving in 2D or 3D space and not undergoing a sudden change in its inertia. Our case study applies to urban tracking through video surveillance cameras, so we can use these filters. We will use Kalman filters in a discrete context. In this context, we will only need 2 images to set up our filter systems N-1 and N. The data from N-1 will allow us to make a prediction of the future position of the points in N and N will allow us to validate the different predictions made. Moreover, the use of Kalman filters remains compatible with the proximity tracking method. We can also use the association effect by proximity to reinforce the tracking by Kalman. In addition to the domains mentioned above, uses of Kalman filters in tracking are possible to allow, for example, in the field of sports to identify a player at any time [16] or in the field of security to fill the gaps in the movement of a person who would pass behind an obstacle for example [24]. However, due to the computational power required to perform both detection and tracking, there is very little work on the subject in the field of urban tracking. Most of the works using Kalman filters in the field of video tracking are directly applied on the latter. The method proposed in this article allows to detach completely from the video part and focuses only on the tracking of anonymous points. In doing so, the work done here can be applied in various domains outside urban tracking. Indeed, this method can be used to track any population of objects as long as they are identifiable by parameters that evolve without abrupt variations.

As for the proximity tracking method, we have the same series of point clouds as input data where each element of the series is associated with a frame of the video. However, since the detection size is important in this method, we will use for each point the information on its height and width in pixels. Thus, we can associate each point with a surface. These point clouds will then be processed independently of each other, but any point cloud will have an impact on the analysis of the following points. To perform the analysis, we consider that we have two types of points: points called “anonymous detected”, because they come directly from the point clouds, and points called “identified by tracking based on the Kalman filter” generated and managed by our Kalman filters. The objective is to associate an “anonymous detected” point with a “Kalman filter-based tracking” point; our “Kalman filter-based tracking” points are managed by the algorithm which will create one of these points as soon as an anonymous point cannot be associated with any existing “Kalman filter-based tracking” point. We end up with a table of points “identified by tracking based on the Kalman filter” which evolves as we receive clouds of points to process.

Consider the following example:-The algorithm receives a point cloud N1 that contains X points (Figure 10a);-The algorithm looks if one of the received “anonymous detected” points shares enough area (Figure 10c) with that of one of the “Kalman filter-based tracking” points (Figure 10b);-If so, the identifier of the “identified by Kalman filter tracking” point is given to the “anonymous detected” point which becomes an identified point (Figure 10d) and the position data of the “anonymous detected” point is integrated with the associated “identified by Kalman filter tracking” point (Figure 10f) so that the trajectory of the “identified by Kalman filter tracking” point continues to evolve in accordance with its “anonymous detected” point (Figure 10e);-Otherwise, a “Kalman filter-based tracking” point is created with the data of the “anonymous detected” point (Figure 10e).

As shown in Figure 11, the processing algorithm using Kalman filters is as follows:

As explained above, the video stream is processed independently of analysis systems, once processed the result is stored as a JSON file. The first step is therefore to convert the data in the JSON file into data that can be processed by the language used (in our case python3). If it is the first data of the file, it is automatically integrated to the known points. The known points in our case are all the points with an identifier. If it is not the first message, the points “identified by Kalman filter” are updated. It is then possible to compare the new position of the points “identified by Kalman filter” and the new points received. Each new received point will be compared to the set of “Kalman filtered” points and will give one of the following results:-The point corresponds to one of the known points;-The point does not correspond to any known point;-One of the points “identified by Kalman filter” does not find a match.

In the first case, the filters of the points “identified by Kalman filters” are updated just as the known points.

In the second case, a new point “identified by Kalman filter” is created in the known points and the known points are updated.

Finally, in the third case, if the filter has remained without a match for too long, it is deleted, otherwise it is left until it finds a match or is deleted.

Once the algorithm has verified each “anonymous detected” point, it assigns a new identifier to points that remained “anonymous detected,” updates the position of the filters, and deletes filters that have remained unidentified for too long or whose coordinates are no longer in the monitored area. The prediction performed by the Kalman filters is based on two functions performing a series of matrix calculations. The first function is an update function that allows the filter to update the information of its known point with the information of the point that has been reassociated with it. The second method is a prediction method which from the last known information will predict information of the future point which could be associated to it. In order to respect the method used by Kalman filters, we have six main matrices:-E, the matrix representing the initial state vector containing the position, velocity, and size information of the point;-A, the transition matrix allowing to go from the state of Image i-1 to I;-O, the observation matrix allowing to define the parameters to be monitored, in our case it is the information known in the point cloud, namely the position and the size;-Q, the covariance matrix representing the process noise vector;-R, the covariance matrix representing the vector of measurement noise;-P, the prediction matrix evolving with each new piece of information;-The signs +, −, · are used for the basics matrixial operation;-The ^(−1)^ is used to invert matrixes.

In the case of the update, we also receive the matrix J representing the vector of new information of the point and which allows reinforcing the prediction to come as well as G an identity matrix of the same dimension as I. In a first step, we update the information of I:(2)$$E=E+((P\cdot O^{T})\cdot {\left(O\cdot \left(P\cdot O^{T}\right)+R\right)}^{\left(-1\right)})\cdot (J-(O\cdot E))$$

This equation allows to update the position information:(3)$$P=(G-((P\cdot O^{T})\cdot {\left(\left(O\cdot \left(P\cdot O^{T}\right)\right)+R\right)}^{\left(-1\right)})\cdot O)\cdot P$$

This equation allows to update the prediction matrix.

In the case of prediction, the calculation is simpler:E = A·E(4)

To predict the future position:(5)$$P=((A\cdot P)\cdot A^{T})+Q$$

This equation allows to calculate the covariance of the error.

Table 3 shows, for a track, its initial state with its coordinates on X and Y in pixels, its velocity on X and Y in pixels/second, and its width and height in pixels. At the end of the initial state, we obtain a prediction for the future position of the track and then the actual state of the track that was associated with the prediction and those for 17 successive images. In the case of Table 3, the results were obtained using Kalman filter tracking, but were also checked manually to be sure that the data extract presented did not show any dropout. This is only an extract, as the original file is more than 3000 images long.

We then see that the predictions are very close to reality. Table 4 shows all the common surfaces between the predictions and the reality. This exercise was performed on several samples in order to validate the efficiency of the tracking on the one hand, but also to determine the threshold from which a prediction can be attached to a real track. The percentage of common surface chosen for the comparison was first arbitrarily fixed at 70%. It was then lowered to 50% because, after several tests, it was found that a rate higher than 50% was indicative of a good match as shown in Table 4 and conversely matches below 50% are mostly overlaps due to individuals crossing or walking close to each other. Moreover, since each point “identified by Kalman filter tracking” can only be associated with one “anonymous detected” point, we find that points “identified by Kalman filter tracking” that are less than 50% overlapping with an “anonymous detected” point have in most cases another “anonymous detected” point on which more than 50% of the surface is common. This effect allows in all cases to select only the matches with more than 50% of the surface and allowed to validate this value.

We thus have two possible cases for each “anonymous detected” point: either the “anonymous detected” point can be associated with a point “identified by tracking based on the Kalman filter”, this association is done if the common surface between the two points is sufficient. In the case where several “anonymous detected” points have a sufficient common surface, it is the point with the largest common surface that will be kept. Either the “anonymous detected” point is not associable with any point “identified by Kalman filter tracking” and in this case a point is created using the position and size information associated with the “anonymous detected” point. Once this check is done, our point cloud is no longer “anonymous detected” and we can wait for the next cloud.

This method also notes cases of stalling when the number of images per second is not sufficient or if the trajectory of an individual is not “natural” enough; the number of stalls is presented in Table 5.

We can see that Table 5 shows better results, because the number of dropouts is less important. We can therefore compare the two methods with the help of these results.

## 5. Discussion and Analysis

All results are presented and compared in Table 6. Table 6 shows in green the method that revealed the least dropouts, in red the one that did the most, and in orange the ties. In general, the most reliable method is the one using Kalman filters. This efficiency can be observed by averaging the dropouts (all videos combined) of each IPS frequency, which shows a minimum of 209% more dropouts with the first method (Table 7).

On the other hand, it was identified that in some video cases the first method would prove more effective. Indeed, the video No. 19 features the raiding of a single person throughout the video. It turns out that the advantage of Kalman filters to be able to anticipate the trajectory of an individual can become a disadvantage if the latter stops abruptly and does not resume his walk. Indeed, the filter allowing to predict the future position of the point, cannot anticipate a sudden stop which has for effect to make the tracking stall. Moreover, the first method consisting in comparing the closest points to connect them will necessarily perform a perfect tracking if there is only one point in each image, because there is no other possibility (Table 8). However, these problems were identified in advance and do not affect the results enough to reveal a real advantage to using the proximity tracking method alone.

The experimental results show that the Kalman filter method performs, in most cases, a much more efficient tracking than the proximity method. This result comes from the integration of a trajectory estimation. This integration in an urban environment is very efficient because it allows us to anticipate the evolution of the position of a person walking normally. However, as soon as the movement of an individual is more random or abrupt, the algorithm will no longer be able to estimate trajectories. The combination of the two methods allows us to be as efficient as possible by having both a short term prediction that generally matches the future image with the prediction made, but also a matching system by proximity. Indeed, in some cases, predictions may not correspond to reality, either because the track has disappeared and in this case, a future prediction should be able to find the track; or because the information used to make the prediction is not necessary, in this case, the association by proximity will allow to find the track and to reinforce the future prediction. Nevertheless, there are still some stalls at the end of the experiments. In most cases, it is a question of stalling due to the passage behind an obstacle or when several people are passing. More rarely, it can happen that the stall is linked to a bad detection and in this case it would be necessary to improve the detection system.

## 6. Conclusions

This paper highlights the advantage of using Kalman filters for human tracking in urban environments. Results are available online for videos used to obtain the data for this paper: https://youtu.be/R-NkqDYEAjM (accessed on 19 July 2021). The proposed methods show that a frame-by-frame tracking can only be efficient if a trace of the past positions is kept. A simple position comparison method already allows obtaining efficient results in a controlled environment: without obstruction and with a controlled number of individuals moving at a distance from each other. The use of this same method with the association of a prediction of future positions makes the tracking system much more efficient and allows it to succeed in tracking individuals in real conditions: with disappearances behind obstacles, crossings of individuals, and abrupt changes of trajectories. However, the system is not infallible, indeed it is possible that the tracking stalls when there are too many individuals in the image that move too quickly. To overcome this problem, several solutions are possible. In this case, the solution implemented is the one presented in [3]. The objective is to focus on a group movement through their evolution in time by studying for example the variation of the standard deviation and the barycenter. This method allows to make much faster conclusions when the number of individuals is too large. Another solution would be the integration of a form factor provided during the video detection and which would be calculated in the same way for all. This value or these values could be calculated using information that is not yet used by the previous methods such as a value related to the color of the detection or to the morphology of the detection for example using the method presented in [5]. This factor, if it evolves smoothly, could even be integrated into the method using Kalman filters and perhaps even used directly in the filter. Another possible solution would be to use the results of the method using Kalman filters to train artificial intelligence through simulations. Indeed, as the method is already working correctly, the results of the latter could allow us to have enough data to train a neural network, for example. Work is in progress to try to improve the method and will be the subject of other publications.

## Figures and Tables

**Figure 1 sensors-21-07234-f001:**
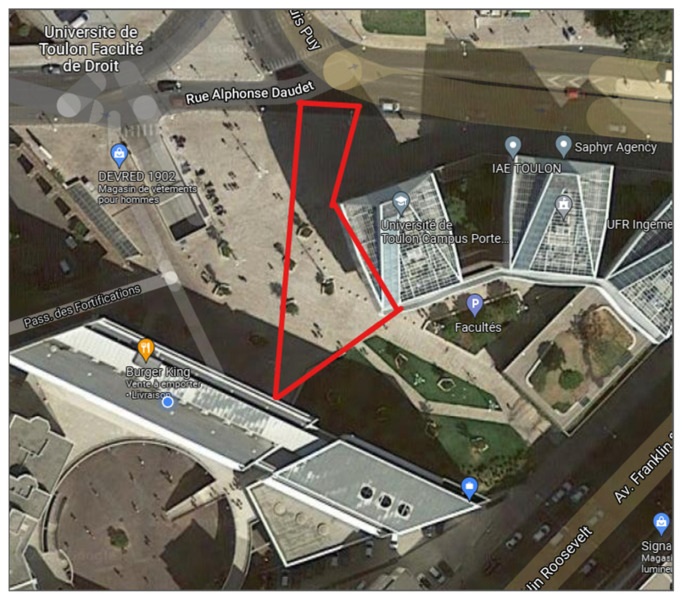
Filmed Area by the study camera, surrounded in red. Location: 43°07′15.5″ N 5°56′21.3″ E.

**Figure 2 sensors-21-07234-f002:**
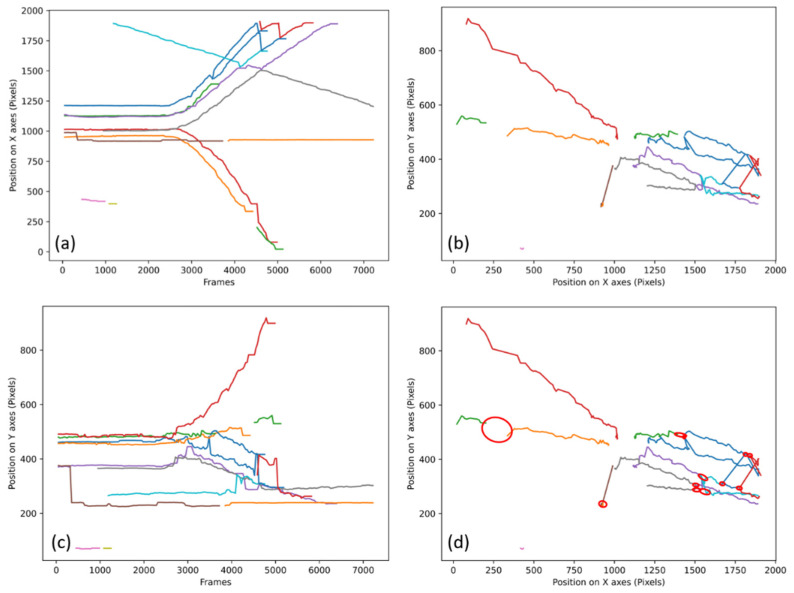
Tracking as a function of time on the x-axis and y-axis, each color represents a tracked point (**a**) Tracking on X of video No. 1 at 30IPS with the first method, (**b**) Tracking on X and Y of video No. 1 at 30IPS with the first method, (**c**) Tracking on Y of video No. 1 at 30IPS with the first method, (**d**) Highlighting of the stalls of video No. 1 at 30IPS with the first method, each stall is circled in red.

**Figure 3 sensors-21-07234-f003:**
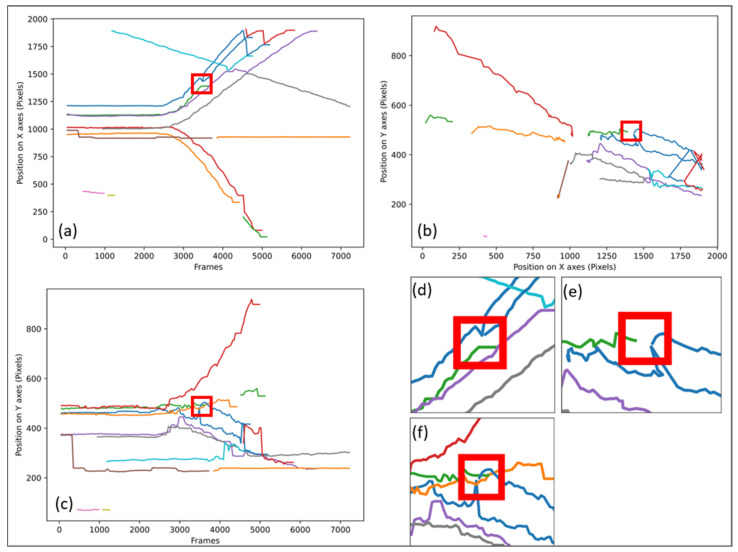
Tracking as a function of time on the x-axis and y-axis, each color represents a tracked point with highlighting of a stall (**a**) Tracking on X of video No. 1 at 30IPS with the first method, (**b**) Tracking on X and Y of video No. 1 at 30IPS with the first method, (**c**) Tracking on Y of video No. 1 at 30IPS with the first method, (**d**) stall highlighting on (**a**), (**e**) stall highlighting on (**b**), (**f**) stall highlighting on (**c**).

**Figure 4 sensors-21-07234-f004:**
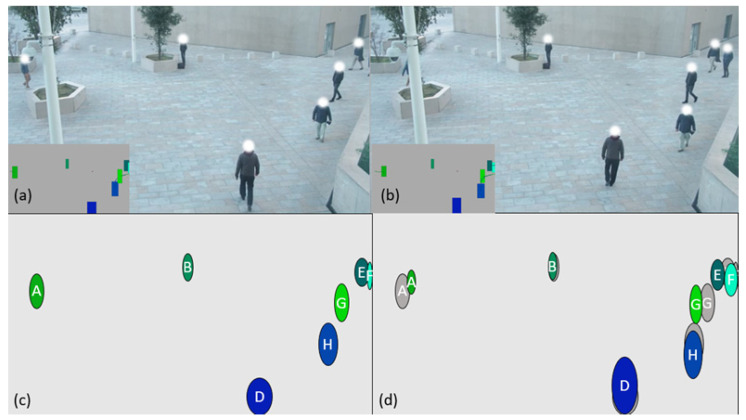
(**a**) graphical representation of the algorithmic view at a time t, (**b**) graphical representation of the algorithmic view at a time t + 1, (**c**) view of the identifiers associated with each point at a time t, (**d**) passage of the identifiers to each new point at a time t + 1.

**Figure 5 sensors-21-07234-f005:**
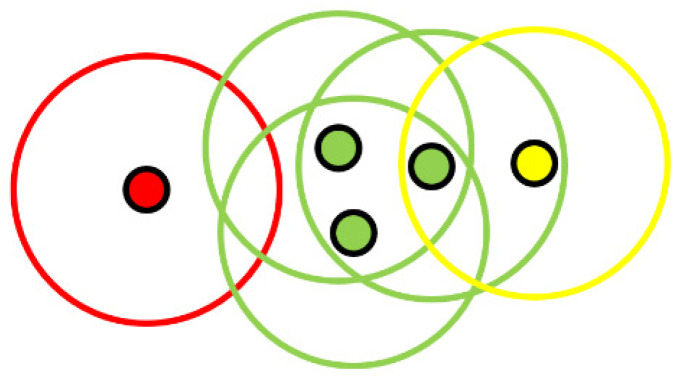
Clustering by DBSCAN: (Green) Clustered points. (Yellow) End points. (Red) Single points.

**Figure 6 sensors-21-07234-f006:**
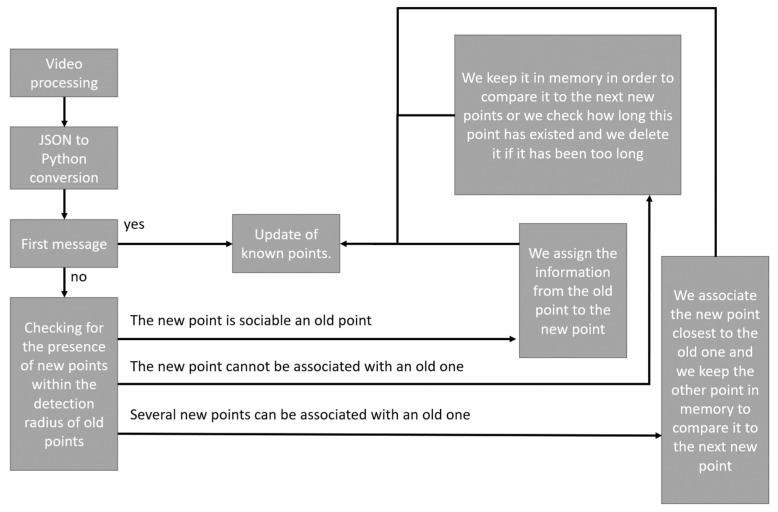
Operation of the proximity tracking algorithm.

**Figure 7 sensors-21-07234-f007:**
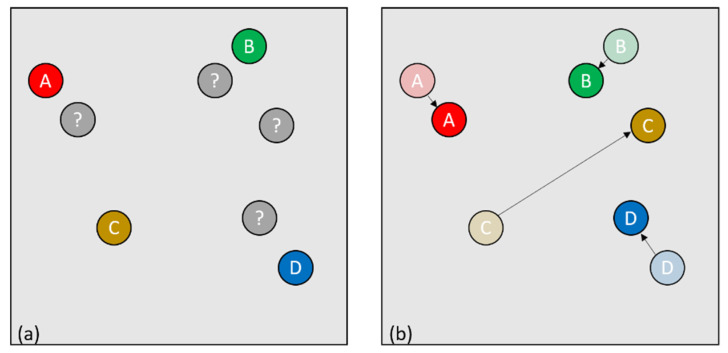
(**a**) Four identified points and four new points received and not identified, one of which disappears and one of which appears elsewhere (**b**) the unidentified points are associated with the identified points.

**Figure 8 sensors-21-07234-f008:**
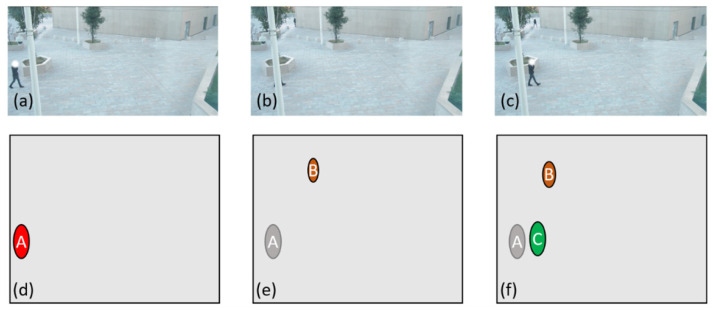
(**a**) an individual enters the image, (**b**) the individual passes behind an obstacle, (**c**) the individual is no longer hidden, (**d**) the point representing the individual in image (**a**) is identified, (**e**) the individual identified in (**d**) is lost, (**f**) the individual identified in (**d**) is still lost and another individual is identified in image (**c**).

**Figure 9 sensors-21-07234-f009:**
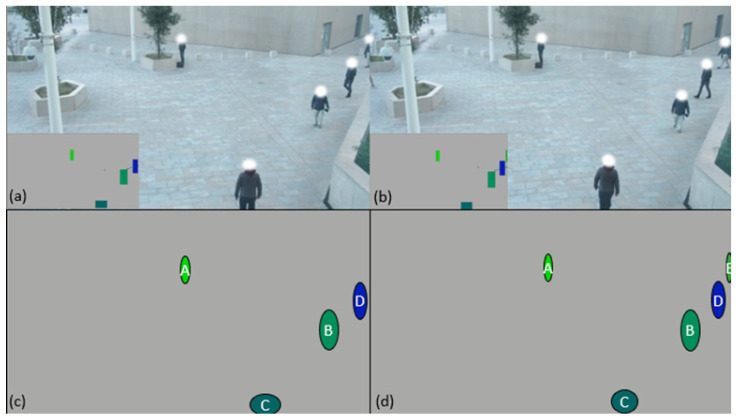
Changeover of four to five individuals in the field of view of the camera. (**a**) Graphical representation of the algorithmic view at a time t with 4 people, (**b**) graphical representation of the algorithmic view at a time t + 1 with 5 people, (**c**) 4 identified points, (**d**) the unidentified points are associated to the identified points the points alone get a new identifier.

**Figure 10 sensors-21-07234-f010:**
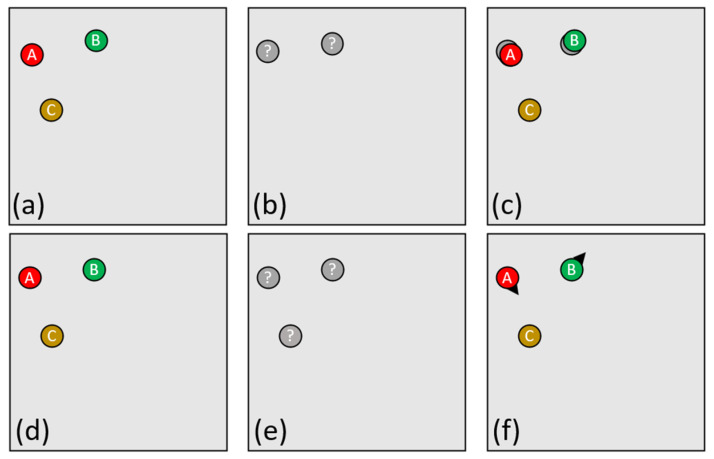
(**a**) three points are identified, (**b**) two points “identified by Kalman filter-based tracking” exist in memory, (**c**) calculation of common surfaces and creation of points “identified by Kalman filter-based tracking” for the points having no pair, (**d**) three points are identified, (**e**) three points “identified by Kalman filter-based tracking” exist in memory, (**f**) the calculation of common surface now allows to know the trajectory of A and B.

**Figure 11 sensors-21-07234-f011:**
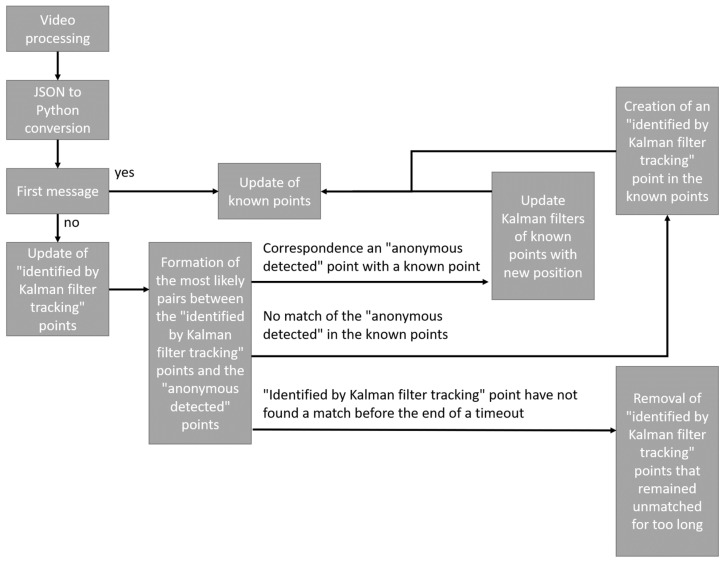
Operation of the tracking algorithm with Kalman filters.

**Table 1 sensors-21-07234-t001:** Characteristic of the videos.

**Video Number**	1	2	5	7	12	17	19
**Obstacles**	3	2	6	2	27	4	1
**Crossings**	2	0	0	1	13	0	0
**Merge**	12	4	6	4	14	4	0
**Number of Individuals**	10	7	9	7	33	9	1
**Video Duration**	7	15	22	18	103	26	115

**Table 2 sensors-21-07234-t002:** Total stalls in proximity tracking on a set of seven videos with a predefined number of IPS.

IPS	Stall						
	Video No. 1	Video No. 2	Video No. 3	Video No. 4	Video No. 5	Video No. 6	Video No. 7
10	12	3	5	6	24	18	0
15	12	2	5	7	18	14	0
20	12	5	5	6	16	14	0
25	12	6	5	7	15	12	0
30	12	7	5	7	22	12	1

**Table 3 sensors-21-07234-t003:** Sample of 17 comparisons between prediction and reality.

	**Initial**	**Prediction No. 1**	**Real No. 1**	**Prediction No. 2**	**Real No. 2**	**Prediction No. 3**	**Real No. 3**	**Prediction No. 4**	**Real No. 4**	**Prediction No. 5**	**Real No. 5**	**Prediction No. 6**
**X Coordinate**	1680	1680	1694	1701	1706	1718	1715	1725	1728	1739	1766	1778
**Y Coordinate**	454	454	449	447	456	463	453	454	445	441	425	419
**X Speed**	0	0	7	7	12	12	10	10	11	11	12	12
**Y SPEED**	0	0	−2	−2	7	7	1	1	−4	−4	−6	−6
**Width**	85	85	92	92	111	111	113	113	111	111	85	85
**Height**	210	210	213	213	218	218	222	222	219	219	201	201
	**Real No. 6**	**Prediction No. 7**	**Real No. 7**	**Prediction No. 8**	**Real No. 8**	**Prediction No. 9**	**Real No. 9**	**Prediction No. 10**	**Real No. 10**	**Prediction No. 11**	**Real No. 11**	**Prediction No. 12**
**X Coordinate**	1780	1793	1787	1787	1795	1804	1811	1823	1820	1830	1831	1841
**Y Coordinate**	422	418	426	426	424	423	420	417	418	415	416	413
**X Speed**	13	13	10	10	9	9	12	12	10	10	10	10
**Y SPEED**	−4	−4	0	0	−1	−1	−3	−3	−3	−3	−3	−3
**Width**	83	83	98	98	109	109	96	96	90	90	87	87
**Height**	207	207	224	224	222	222	216	216	207	207	200	200
	**Real No. 12**	**Prediction No. 13**	**Real No. 13**	**Prediction No. 14**	**Real No. 14**	**Prediction No. 15**	**Real No. 15**	**Prediction No. 16**	**Real No. 16**	**Prediction No. 17**	**Real No. 17**	
**X Coordinate**	1840	1850	1850	1860	1855	1862	1862	1869	1870	1877	1875	
**Y Coordinate**	414	411	409	405	406	402	402	398	398	394	396	
**X Speed**	10	10	10	10	7	7	7	7	7	7	6	
**Y Speed**	−3	−3	−4	−4	−4	−4	−4	−4	−4	−4	−3	
**Width**	8	89	91	91	89	89	87	87	85	85	81	
**Height**	198	198	197	197	204	204	198	198	199	199	200	

**Table 4 sensors-21-07234-t004:** Common area between reality and prediction on a sample of 17 images.

	**Image No. 1**	**Image No. 2**	**Image No. 3**	**Image No. 4**	**Image No. 5**	**Image No. 6**	**Image No. 7**	**Image No. 8**	**Image No. 9**
**Common Area in Pixels**	14,768	17,748	23,320	23,100	15,540	16,434	16,517	19,800	20,448
**Prediction + Actual Area**	22,678	26,046	25,964	26,295	25,854	17,832	22,616	26,350	24,486
**Common Area in %**	65.12	68.14	89.82	87.85	60.11	92.16	73.03	75.14	83.51
	**Image No. 10**	**Image No. 11**	**Image No. 12**	**Image No. 13**	**Image No. 14**	**Image No. 15**	**Image No. 16**	**Image No. 17**	
**Common Area in Pixels**	18,009	17,400	17,226	17,355	16,464	17,226	16,830	15,563	
**Prediction + Actual Area**	21,357	18,630	17,796	18,194	19,619	18,156	17,311	17,552	
**Common Area in %**	84.32	93.40	96.80	95.39	83.92	94.88	97.22	88.67	

**Table 5 sensors-21-07234-t005:** Total dropouts in tracking with Kalman filter on a set of seven videos with a predefined number of IPS.

IPS	Stall						
	Video No. 1	Video No. 2	Video No. 3	Video No. 4	Video No. 5	Video No. 6	Video No. 7
10	8	2	2	2	4	3	1
15	4	0	2	2	2	1	2
20	3	2	3	2	4	0	1
25	3	2	3	1	3	0	1
30	1	3	2	3	4	0	1

**Table 6 sensors-21-07234-t006:** Comparison of the number of stalls between the two versions of the algorithm.

IPS	Stall V1							Stall V2						
	1	2	3	4	5	6	7	1	2	3	4	5	6	7
10	12	3	5	6	24	18	0	8	2	2	2	4	3	1
15	12	2	5	7	18	14	0	4	0	2	2	2	1	2
20	12	5	5	6	16	14	0	3	2	3	2	4	0	1
25	12	6	5	7	15	12	0	3	2	3	1	3	0	1
30	12	7	5	7	22	12	1	1	3	2	3	4	0	1

**Table 7 sensors-21-07234-t007:** Comparison of dropouts at variable IPS frequency, all videos combined.

IPS	Average Error IPS with Tracking by Proximity	Average Error IPS with Kalman Filter Tracking	Difference Tracking by Proximity on Kalman Filter Tracking
10	10	3	+209.09%
15	8	2	+346.15%
20	8	2	+286.67%
25	8	2	+338.46%
30	9	2	+371.43%

**Table 8 sensors-21-07234-t008:** Comparison of dropouts on all videos with all IPS.

Video No.	Average Error with Tracking by Proximity	Average Error with Kalman Filter Tracking	Difference Tracking by Proximity on Kalman Filter Tracking
1	12	4	+215.79%
2	5	2	+155.56%
3	5	2	+108.33%
4	7	2	+230.00%
5	19	3	+458.82%
6	14	1	+1650.00
7	0	1	−83.33%

## Data Availability

Not applicable.

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
