# Peer review of "Behavioral Analysis and Individual Tracking Based on Kalman Filter: Application in an Urban Environment"

_sensors, 2021, doi:10.3390/s21217234_

Round 1
Reviewer 1 Report
The paper discusses a tracking method through two approaches: tracking by proximity and using Kalman filters. While the topic is interesting for the community some issues prevent the publication of this contribution in the present form.
1) My main concern is related to the novelty of the work. It is not clear if the proposed approach with the Kalman filters is new or if it has already been used for the proposed or similar applications. Please better clarify how this contribution represent a
2) The set-up used to acquire the dataset has not been described clearly. What are the camera characteristics and how they impact the tracking ability of the two considered algorithms?
3) In fig.1 and 2 legends are missing and the graphs are difficult to understand.
4) Please consider putting the figures' caption under the corresponding figures and not in a new page. Also, the tables should be placed in only one page and not split in two pages.
Author Response
Thank you for the interest and constructive feedback you have shown in reviewing this article.
We have dealt with all the remarks that were made to the best of our ability. We hope that we have taken into account all of your comments and have responded in the expected manner. Below you will find all the changes that have been made in response. The line numbers are those that appear when the document is in tracked changes without any markings.
Regarding the remark: "My main concern is related to the novelty of the work. It is not clear if the proposed approach with the Kalman filters is new or if it has already been used for the proposed or similar applications. Please better clarify how this contribution represent a"
Several points have been clarified to show the scientific novelty of the work. Several already existing methods have been re-used in a different way and present in their use a novelty. In order to answer this remark more clearly, you can consult lines 218 to 238 which explain a different exploitation of the DBSCAN clustering method. You can also consult line 361 to 371 which explains how the use of Kalman filters in this article differs from the use of Kalman filters in the literature. You will find, in addition to these elements, two new figures and their explanations on line 239 to 262 and on line 406 to 427 which present the functioning of the implemented algorithm which also presents the way in which the contributions have been exploited. Additionally, two new figures have been added namely figures 6 and 11 to illustrate the explanation..
To answer the remark: "The set-up used to acquire the dataset has not been described clearly. What are the camera characteristics and how they impact the tracking ability of the two considered algorithms?"
Elements have been added as to how the videos were obtained. From line 95 to line 108, you will find information on the shooting conditions and on the tools used. Regarding their impact on the tracking ability of the two algorithms, we have tried to emphasize again that the video remains the basic element of the analysis, but that the processing is however not dependent on the quality of the video.
Regarding the remark about the figures, "In fig.1 and 2 legends are missing and the graphs are difficult to understand."
The explanation text for these has been reworked to be more understandable in line 153 and 160. These figures show the movement of individuals either as a function of time or distance. The values and the colors are of little importance, what is emphasized and what is important to see is the break in the continuity of the different lines. We hope that the explanations added in the text will allow a better interpretation.
Finally for the remark : "Please consider putting the figures' caption under the corresponding figures and not in a new page. Also, the tables should be placed in only one page and not split in two pages."
This is indeed a layout error, it has been corrected. We also noticed other small errors of the same kind and corrected them as well.
Reviewer 2 Report
Comments and suggestions for the reviewed work are included in the attachment.

Author Response
Thank you for the interest and constructive feedback you have shown in reviewing this article.
We have dealt with all the remarks that were made to the best of our ability. We hope that we have taken into account all of your comments and have responded in the expected manner. Below you will find all the changes that have been made in response. The line numbers are those that appear when the document is in tracked changes without any markings.
We have understood all the remarks related to the conditions of realization of the various experiments. We have therefore tried to provide as much information as possible throughout the document on how the different data sets were obtained. However, we would like to emphasize that in the context of this article, the videos and their qualities only play a secondary role. Indeed, the purpose of the algorithms is to work on anonymous point clouds. We have therefore included the explanations related to this point in the hope that it will bring you satisfaction.
We also tried to take into account your remarks on the meaning of the signs used in the matrix calculations. It turns out that one of the elements you mentioned, namely "S", was a part of the equation that we had forgotten to include in the formula. We hope that the corrections made will allow you to better appreciate the article.
Regarding the remarks: "what programming tools / experimental environment were used to process the data?"
We have added line 117 to 146 which provides more information about the data format used and the development tools. We had few constraints on the choice of tools, so the most logical choices were those that allowed us to move forward most quickly. It so happens that the language chosen was Python 3 and in general the tools were selected either following the state of the art or because we were already aware of their efficiency.
About the remark: "what kind of Kalman filter model (ordinary, extended, etc.) was adopted and what its parameters were?"
We have tried to provide throughout the paper new information about the Kalman filters used. We will use Kalman filters in a discrete context, but you will find in particular in the part "4. Kalman filter tracking", some additional information on the types of filters as well as more precise explanations on the way the filter was used within the algorithm.
Regarding the remark: "what shape and size of time window (frames and possibly their weights) was used to generate trajectory local prediction? "
The information you refer to was not instrumental in the trajectory prediction however we have tried to fill this information gap in the document by providing a data format that was used as well as other information on the content of the processed data. You should find all the necessary information in what has been added between lines 117 and 146 and in the explanations found between lines 239 and 262 and the explanation between lines 406 and 427 where two new figures have been added namely figures 6 and 11.
For the remark: "exactly what kind of classic tracking algorithms were used in comparison to the Kalman filter method, their parameters? models? "
As the document deals with two different methods, we have added in the same way as for the Kalman filter method the information on the functioning of the algorithm using proximity. You will find more explanations on how the groupings are made in the proximity method and on the functioning of the algorithm between lines 218 and 262.
Finally, concerning the remark: "can the authors provide a comparison of their approach to the KLT method used in optical flow methods?"
The days of revisions have allowed us to try to understand and grasp the method to which you refer. However, the level of knowledge of this method not being sufficient to obtain an interesting comparison between the 2 methods this point could not be completely approached. However, a part has been added in the state of the art between lines 59 and 64 highlighting the interest of this method which at first sight is mainly applied to videos, but which consequently only partially answers our problem of data anonymization. However, I thank you for having made us discover this method that we had not had the opportunity to exploit.
Regards,
Amaury AUGUSTE.
Reviewer 3 Report
The article is interesting and correctly drafted but a few notes need to be included, namely:
- Describe the algorithm used in more detail.
- Add more details about the type of Kalman filter use
- Provide more detail on where the data were collected.
- If possible please give a mathematical model of the developed algorithm.
- Conduct a more in-depth analysis of the state of the art (cite more works from the area researched).
Author Response
Thank you for the interest and constructive feedback you have shown in reviewing this article.
We have dealt with all the remarks that were made to the best of our ability. We hope that we have taken into account all of your comments and have responded in the expected manner. Below you will find all the changes that have been made in response. The line numbers are those that appear when the document is in tracked changes without any markings.
Regarding the remark: "Add more details about the type of Kalman filter use"
As for the previous remark, you will find explanations about the use of Kalman filters in lines 406 to 427 and the matrix calculations have also been corrected and completed. We hope that these complements will allow you to obtain the desired details and to add more precision throughout the document.
Regarding the remarks: "Describe the algorithm used in more detail."
And "If possible please give a mathematical model of the developed algorithm."
We do not have a mathematical model of the developed algorithm, however, you should find all the information about how the algorithms work at an accessible level of understanding through the two algorithm explanation charts. You will find more details about the functioning of the algorithms in lines 218 to 262 and in lines 406 to 427 where two new figures have been added namely figures 6 and 11.
Regarding the remark: "Provide more detail on where the data were collected."
More details have been added in the "2. database" section on how the different videos were obtained, but also where they were shot with a new figure (fig.1). The objective being to have a total control of what happens on the video, we are in charge of making our own videos trying to get as close as possible to the parameters of the video surveillance cameras used in the project.
Finally, for the remark: "Conduct a more in-depth analysis of the state of the art (cite more works from the area researched)."
We have added several other references in the paper on similar technologies and on the use of Kalman filters in related areas. However, the lack of citations is also due to a lack of literature. Kalman filters are generally used for prediction, the use of Kalman filters in the field of video surveillance is possible, but the way we exploit this method is uncommon, that is why there are few references available. On the other hand, you should find in the two new references added elements allowing you to complete the understanding of the method used in this article.
Regards,
Amaury AUGUSTE.
Round 2
Reviewer 1 Report
While the authors clarified the methods that give a more deep insight in their work the novelty of the work compared to the literature is not clear. The authors should state both in the abstract and in the introduction how their work differs from past and existing projects.
Author Response
Thank you for your feedback,
We understand your remark and we have tried to be clearer on the point that you expose. We have added in the abstract, how the use of Kalman filters is new compared to the use that can be found in the current literature. You will find these changes in line 13 to 15 and 18.
In the same way, we have also added another paragraph in the introduction which emphasizes the novelty of the method, especially in the field of tracking and the detection of abnormal behaviors through the parameters that the tracking brings. You will find this modification in lines 88 to 99.
We hope that the addition of these elements, in addition to the explanations already given in the first review, will suit you.
Wishing you a good reception.
Reviewer 2 Report
In the present paper form, the authors have taken into account most of the suggestions and comments in the opinionated manuscript, therefore, in my view, it can be submitted for publication.
Author Response
Thank you for your feedback and for your time,
We hope that we have taken into account all the feedback from other reviewers and have met all your expectations.